# Pulmonary Conventional Type 1 Langerin-Expressing Dendritic Cells Play a Role in Impairing Early Protective Immune Response against *Cryptococcus neoformans* Infection in Mice

**DOI:** 10.3390/jof8080792

**Published:** 2022-07-28

**Authors:** Lorena Guasconi, Ignacio Beccacece, Ximena Volpini, Verónica L. Burstein, Cristian J. Mena, Leonardo Silvane, Mariel A. Almeida, Melina Mara Musri, Laura Cervi, Laura S. Chiapello

**Affiliations:** 1Departamento de Bioquímica Clínica, Facultad de Ciencias Químicas, Universidad Nacional de Córdoba, Córdoba X5000HUA, Argentina; lguasconi@unc.edu.ar (L.G.); ibeccacece@unc.edu.ar (I.B.); ximenavolpini@unc.edu.ar (X.V.); vburstein@unc.edu.ar (V.L.B.); cristian.mena@unc.edu.ar (C.J.M.); leonardo.silvane@unc.edu.ar (L.S.); marielalmeida96@gmail.com (M.A.A.); 2Centro de Investigaciones en Bioquímica Clínica e Inmunología (CIBICI), Consejo Nacional de Investigaciones Científicas y Técnicas (CONICET), Córdoba X5000HUA, Argentina; 3Instituto de Investigación Médica Mercedes y Martín Ferreyra (INIMEC), Consejo Nacional de Investigaciones Científicas y Técnicas (CONICET), Córdoba X5016GCA, Argentina; mmusri@immf.uncor.edu; 4Departamento de Fisiología, Facultad de Ciencias Exactas, Físicas y Naturales (FCEFyN), Universidad Nacional de Córdoba, Córdoba X5016GCA, Argentina

**Keywords:** cryptococcosis, lung dendritic cells, DC1

## Abstract

Lung dendritic cells (DC) are powerful antigen-presenting cells constituted by various subpopulations that differ in terms of their function and origin and differentially regulate cell-mediated antifungal immunity. The lung is the primary target organ of *Cryptococcus neoformans* and *C. gattii* infections, which makes it essential in the establishment of the first line of anti-cryptococcal defense. However, the lung-specific dynamics and function of DC subsets are poorly understood in cryptococcosis. In this study, we provide evidence for the in vivo function of a conventional langerin-expressing DC1 dendritic cell (LangDC1) population during the first week of intratracheal *C. neoformans* infection in mice. By using conditional depletion of LangDC1 after diphtheria toxin treatment of LangDTREGFP mice, we demonstrate that these animals better control the fungal infection and produce type 1 and 17 cytokines in the context of a type 2 immune response, favoring a predominance of iNOS over arginase-1 expression by pulmonary cells. Our results suggest that LangDC1 cells play a role in impairing immune response for the clearance of *C. neoformans* in the early stage of pulmonary infection.

## 1. Introduction

Cryptococcosis is a life-threatening fungal infection caused by the inhalation of encapsulated yeasts of *Cryptococcus neoformans* and *C. gattii*, acquired from the environment. *C. neoformans* is the etiological agent in more than 90% of cases of cryptococcosis worldwide, causing fatal meningoencephalitis predominantly in patients with HIV [1].

It is widely accepted that type 1 cell-mediated immunity, involving adaptive IFN-γ-producing type 1 T helper (Th1) or innate (ILC1, Tγδ, NK or NKT) cells, and, to a lesser degree, type 17 (IL-17-producing) lymphocytes are critical for controlling cryptococcosis. Type 1 and type 17 inflammation promote antifungal defenses by classical macrophage activation, which eventually kills the yeasts and limits extrapulmonary fungal dissemination. In contrast, susceptible hosts are unable to control the infection due to an unbalanced type 2 immune response, mediated by IL-4, IL-13 and IL-5 cytokines released by Th2 lymphocytes or innate lymphoid cells (ILC2). Type 2 immune response promotes the activation of alternative macrophages, which are permissive to intracellular *C. neoformans* proliferation [2].

The lungs play a central role in the first line of anti-cryptococcal defense, although the tissue-specific dynamics and function of the immune system in this organ are poorly understood. Dendritic cells (DC) are unique professional antigen-presenting cells that initiate or modulate immune response. Remarkably, a variety of DC subsets have been described in terms of their function and origin that differentially regulate the function of T lymphocytes. Nonetheless, the basic mechanisms by which pulmonary DC subpopulations influence immunity to fungal pathogens remain scarcely investigated [3,4].

Although the heterogeneity of DC in the lungs is still being uncovered, they broadly fit into two main subsets: plasmacytoid DC (pDC) and myeloid or conventional DC (cDC), with the latter being further subdivided into two distinct subsets, cDC1 and cDC2, based on their ontogeny and distinct functional specialization [5,6]. The cDC1 (or DC1) are closely associated with the lamina propria of the airway epithelium and are distinguished by the expression of CD103 (CD11c+ CD103+ CD11b−), whereas cDC2 (or DC2) lack CD103 but express CD11b (CD11c+ CD103− CD11b+) and are mainly found in the lung interstitium [7]. Furthermore, CD207 (langerin), a marker of skin Langerhans cells, is exclusively expressed by a subpopulation of pulmonary DC1 (langerin+ CD103+ CD11c+ DC) [8,9,10]. These langerin-expressing DC1 (LangDC1) are tissue-resident cells specialized in sampling antigens from the alveolar space and, like all DC subsets, are able to migrate and transport antigens to draining lymph nodes [7]. CD103+ DC1 form a highly developed network in the epithelial layer of murine airways and show long protrusions between the basolateral space of basal epithelial cells. Some studies call them Langerhans-type dendritic cells, but unlike epidermal LC (yolk sac and fetal liver origin), pulmonary CD103+ lang+ DC derive from a myeloid precursor in the bone marrow and are related to the CD8+ of the lymphoid organs. Lung DC1 have been shown to preferentially acquire particulate antigens or apoptotic cells, stimulating T cell responses by antigen cross-presentation or inducing tolerance by regulatory T cells [3,5,7,8,9]. Experimental data from cryptococcal infection in mice have shown that DC2 infiltrate lung parenchyma, promote fungal clearance and elicit memory-like DC response against *C. neoformans* [4,11,12,13,14,15]. In contrast, although DC1 have been described in the lung tissue during cryptococcal infections [16,17], the role of this pulmonary DC1 population remains unclear.

In this study, we provide evidence for the in vivo function of pulmonary LangDC1 in *C. neoformans*-infected *Lang-DTREGFP* mice. These C57BL/6 mutant mice express the human diphtheria toxin receptor (DTR) and *EGFP* genes downstream of the internal stop codon of the langerin (*CD207*) gene, and they are a useful tool for both visualizing and specifically ablating LangDC1 after administration of diphtheria toxin (DT) [18]. In this sense, conditional depletion of lung DC1 has been utilized to study their role in antiviral immunity, allergic response and tolerance induction [9,19,20,21]. Our data further demonstrate the contribution of LangDC1 to impairing protective immunity in the early stage of experimental *C. neoformans* pulmonary infection.

## 2. Materials and Methods

### 2.1. Mice

C57BL/6 wild type (WT) mice were obtained from the Universidad Nacional de La Plata (La Plata, Argentina). *Langerin-DTREGFP* (*LangDTREGFP*) transgenic mice were kindly provided by Bernard Malissen (INSERM, Paris, France). Mice were housed in the Animal Facility of CIBICI-CONICET, Facultad de Ciencias Químicas, Universidad Nacional de Córdoba (Córdoba, Argentina) under international guidelines. All experimental protocols were approved by our Institution’s Ethics Committee (Dean’s Resolution no. 785/18).

### 2.2. Cryptococcus neoformans

*Cryptococcus neoformans* strain 52D (Universidad Nacional de Córdoba’s stock culture collection) was used. Living yeasts of *C. neoformans* were grown in Sabouraud medium for 48–72 h at 37 °C. The cultures were then washed with nonpyrogenic saline, counted and diluted to 2 × 10^5^ CFU (colony-forming units)/mL in sterile nonpyrogenic saline.

### 2.3. Pulmonary Infection and Langerin-Expressing Cell Depletion in LangEGFPDTR Mice

Intratracheal inoculation was carried out as described by Revelli et al. [22]. Briefly, mice were anesthetized by intraperitoneal injection of ketamine/xylazine (100/10 mg/kg body weight). For inoculation, animals were suspended on a plexiglass backboard in supine position by their incisors. The mouth was opened, the tongue was extended out with forceps and a needle was used to gently reposition the soft palate, expose the tracheal opening and dispense 100 ul of inoculum (10^4^ CFU) between the vocal cords.

To deplete langerin+/DTR cells, diphtheria toxin (DT, Merck, Darmstadt, Germany) was applied 48 h before infection, intranasally (i.n., 50 ng) and intraperitoneally (i.p., 400 ng) [23]. Untreated control mice were injected with saline solution.

### 2.4. Histology

Lungs were fixed in 10% neutral buffered formalin for 48 h, followed by paraffin embedding. Then, sections of 5 µm were stained with hematoxylin and eosin (H&E). Infiltrating leukocyte percentages were determined by cell counting according to microscopy morphology as eosinophils, neutrophils or mononuclear cells.

### 2.5. Fungal Burden 

Lungs were removed, weighed and then homogenized in a physiological solution with 50 mg/mL of gentamicin, under sterile conditions. An aliquot (100 µL) of the homogenates from each animal was then inoculated onto Sabouraud glucose agar to determine the CFU per gram of tissue after 3 days of incubation at 30 °C.

### 2.6. Cultures of Pulmonary Cells

Lungs were perfused in situ via the right heart with phosphate-buffered saline (PBS). Subsequently, the lungs were excised, washed in PBS, minced and enzymatically digested to a single-cell suspension. For enzymatic digestion, lung tissue was incubated at 37 °C for 45 min in a six-well culture dish containing 7 mL of RPMI medium with Liberase TM (Roche, Mannheim, Germany) (50 μg/mL) and DNase I (Roche, 1 μg/mL). Then, cell homogenates were filtered over a 40 μm cell strainer and centrifuged at 500× *g* for 5 min at 4 °C. Finally, total numbers of viable lung cells were prepared for culture or flow cytometry analysis [24,25]. Lung cells were cultured in RPMI supplemented with 10% fetal calf serum (FCS, Natocor, Córdoba, Argentina) and 50 µg/mL gentamicin, for 24 h in 96-well plates (1 × 10^6^ cells/well) at 37 °C in an atmosphere of 5% CO_2_, without additional stimuli.

### 2.7. Flow Cytometry Analysis

To determine the efficiency of LangDC1 depletion or to analyze lung immune cell populations from untreated or DT-treated *LangDTREGFP* mice, lung cells were stained with fluorochrome-labeled aqua dead viability marker (Aqua Zombie, BioLegend, San Diego, CA, USA, cat. N° 423101), anti-CD45 (eBioscienceInvitrogen, ThermoFisher Scientific, Buenos Aires, Argentina, Cat.N° 45-0454-80, clone 104), anti-CD11c (Biolegend, cat N°117310, clone N418), anti-CD103 (Biolegend, cat N° 1214032, clone 2-E7), anti-F4/80 (eBioscience, Invitrogen, cat N° 11-4801-81, clone BM8), anti-CD11b (BD, Buenos Aires, Argentina, cat N° 557397, clone M1/70), anti-Ly6G (BD, N° 560599, clone 1A8) or anti-CD3 (BD, N° 557030, clone 1F4) antibodies, and analyzed by flow cytometry (BD FACSCanto, BD Biosciences, CA, USA). Autofluorescence was assessed using untreated cells and control isotypes.

### 2.8. Cytokine Production

Supernatants of lung cell cultures were collected at 24 h and assayed for TNF, IL-10, IL-4, IFN-γ, IL-12p70 (BD Bioscience, San Diego, CA, USA), IL-13 (Biosource, Camarillo, CA, USA) and IL-17AF (e-Bioscience, Invitrogen, ThermoFisher scientific, San Diego, CA, USA) production by sandwich ELISA according to the manufacturers’ instructions supplied with the cytokine-specific kits.

### 2.9. Fungal Growth Inhibition by Lung Cells

Lung cells were obtained by enzymatic digestion as described above, resuspended in RPMI supplemented with 10% FCS and 50 µg/mL of gentamicin and cultured for 90 min in 24-well plates (5 × 10^5^ cells/well). Non-adherent cells were rinsed away, and the remaining adherent cells were cultured with a suspension of 1 × 10^3^ yeasts of *C. neoformans* for 48 h [25]. To determine fungal growth inhibition, CFU were quantified after culture. A volume of distilled water was added to each well, and 100 µL of a 200-fold dilution of each sample was inoculated in Sabouraud dextrose agar plates. After 48 h, CFU ratio was calculated by dividing the final CFU number by the initial number of yeasts in the suspension.

### 2.10. Arginase Activity Assay

Adherent lung cells were washed and lysed for arginase activity determination as described by Corraliza et al. [26]. Briefly, after lysis of cells in Triton X-100 containing 5 µg pepstatin, 5 µg aprotinin and 5 µg antipain as protease inhibitors, the mixture was stirred for 30 min at room temperature. Then, 50 µL of 10 mM MnCl_2_ and 50 mM Tris–HCl were added to lysed cells to activate the enzyme with heating for 10 min at 56 °C. The arginine hydrolysis was initiated by the addition of 25 µL 0.5 M L-arginine, pH 9.7, followed by heating at 37 °C for 45 min. The reaction was stopped with 400 µL H_2_SO_4_ (96%)/H_3_PO_4_ (85%)/H_2_O (1/3/7, *v*/*v*/*v*). Then, after the addition of 25 µL α-isonitrosopropiophenone, followed by heating at 95 °C for 45 min, the urea concentration was measured at 540 nm using a spectrophotometer. Proteins were measured by the Bradford method, and the results were expressed as µg of urea per µg of protein.

### 2.11. Immunofluorescence

Microsections from paraffin-embedded lungs of *C. neoformans*-infected *Lang-DTREGFP* mice (7 dpi), untreated or DT-treated, were permeabilized and blocked with 0.2% Triton X-100/5% bovine serum albumin (BSA)/PBS for 1 h at room temperature. Then, slides were stained with antibodies to Arginase-1 (14-977982-82, eBioscience, ThermoFisher, Buenos Aires, Argentina) or inducible nitric oxide synthase (iNOS) (ab15323, Abcam, Boston, MA, USA). Anti-rabbit antibodies labeled with fluorochromes (ThermoFisher) were used as secondary antibodies. The nuclei were stained with DAPI (Invitrogen). Series of z-stack images of 0.5 μm thickness were acquired using an Olympus FV1200 laser scanning confocal microscope (Olympus Corporation, Tokyo, Japan). Image analysis was carried out using the software FI-JI/ImageJ.

### 2.12. Statistical Analysis

Data are expressed as mean ± SEM, and all experiments were performed two or three times. Six mice per group were used. Statistical differences were calculated with Student’s *t*-test or ANOVA. A *p*-value of 0.05 was considered significant.

## 3. Results

### 3.1. LangDC1 Depletion Favors the Pulmonary Clearance of C. neoformans at the Early Stage of Infection

Several studies have demonstrated that DC are key to sensing and phagocytosing *C. neoformans* as well as to orchestrating the adaptive anti-cryptococcal immune responses [16]. However, the role of the pulmonary DC1 subpopulations in cryptococcosis remains unexplored. Langerin-expressing DC1 (LangDC1) can be selectively studied by taking advantage of the expression of langerin (CD207) in *LangDTREGFP* transgenic mice and conditionally ablating langerin-expressing cells with DT injection [20,23]. Taking into account that langerin is expressed in a proportion of pulmonary DC1 (CD11c+ CD103+), we evaluated the efficiency of the LangDC1 depletion by observing langerin-expressing cells (EGFP+) in lung cryosections from untreated or DT-treated *LangDTREGFP* mice (Figure 1A). After 2 days of intranasal and intraperitoneal DT administration, a complete depletion of langerin-expressing cells was observed in the lung sections from *LangDTREGFP* mice, which recovered about 50% of EGFP+ cells at 7 days post DT treatment (Figure 1B). Lung sections from DT-treated *LangDTREGFP* mice showed a similar fluorescent background to those from WT mice. Consistent with the fact that langerin is exclusively expressed on pulmonary CD103+ CD11c+ DC1 [22], flow cytometry analysis of lung cell suspensions also displayed a significant decrease in the frequency of CD103 expression on the CD45+ CD11c+ DC gated population in DT-treated *LangDTREGFP* mice compared with those from untreated control animals (Figure 1C).

On the basis of these results, we further evaluated the susceptibility to pulmonary infection by histopathology analysis and lung fungal burden in LangDC1-depleted (*LangDTREGFP*+DT) and LangDC1-competent (untreated *LangDTREGFP*) mice, after 7 days of intratracheal infection with *C. neoformans*. As previously reported in *C. neoformans*-infected C57BL/6 mice [27,28], *LangDTREGFP* animals revealed a lung histopathology consistent with allergic bronchopulmonary mycosis, with numerous encapsulated yeasts surrounded by an inflammatory infiltrate with predominance of neutrophils and some eosinophils in the lung parenchyma (Figure 2A). In contrast, LangDC1-depleted mice showed scarce yeasts and inflammation in lung tissue, with some intracellular *C. neoformans* in multinucleated giant cells (MGC). Moreover, these LangDC1-depleted animals had a lung architecture similar to uninfected controls (Figure 2A). In line with histological data, LangDC1-depleted animals revealed a significant reduction in colony-forming units (CFU) in lung homogenates compared with those from untreated control mice (Figure 2B). In contrast, no differences in lung fungal burden were observed in DT-treated or untreated WT mice (data not shown). Microscopic quantification of infiltrating leukocytes also revealed that LangDC1-depleted mice have a significant reduction in neutrophils and eosinophils, with increased recruitment of mononuclear cells into the infected tissue (Figure 2C). Accordingly, flow cytometry analysis of lung cell suspensions showed lower frequency of CD11b+ (myeloid cells), F4/80+ (macrophages) and Ly6G+ (neutrophils) with a higher percentage of CD3+ lymphocytes in live CD45-expressing cells from LangDC1-depleted mice compared with those from LangDC1-competent animals (Appendix A).

These results demonstrate that the depletion of LangDC1 favors a pulmonary protective immune response against *C. neoformans* infection, suggesting a role of LangDC1 in suppressing the anti-cryptococcal mechanisms in the lung.

### 3.2. LangDC1 Downregulate Cytokine Production and Antifungal Activity of Lung Cells during the Early Stage of Cryptococcal Infection

It is widely accepted that type 1 and type 17 cell-mediated immunity plays a crucial role in host protection against cryptococcal infection while type 2 immune response has been associated with detrimental effects [29,30,31]. Dendritic cells are professional antigen-presenting cells specialized in initiating or modulating cellular immune response profiles against pathogens; however, it is currently unclear how DC1 subpopulations drive the antifungal immunity to *C. neoformans* in vivo. To explore the effects of LangDC1 depletion on cell-mediated immunity, cytokine production by lung cells from 7-day-infected mice was determined after 24 h culture. Figure 3A shows that ex vivo pulmonary cells from LangDC1-competent mice (*LangDTREGFP*) produced type 2 (IL-4, IL-13) and IL-10 cytokines in culture supernatants, without detectable levels of type 1 or type 17 cytokines. Nevertheless, cells from LangDC1-depleted mice (*LangDTREGFP*+DT) displayed a significant increase in type 2 cytokine synthesis in comparison to those from LangDC1-competent mice, as well as production of type 1 (IFN-γ, IL-12) and type 17 (IL-17A) cytokines. These results demonstrate that in the absence of LangDC1 there is a shift of cytokine synthesis by lung cells after *C. neoformans* infection, with production of type 1 and type 17 cytokines in an exacerbated type 2 environment.

On the other hand, macrophages are central effector leukocytes against *Cryptococcus neoformans*, playing a major role in the pathogenesis of the disease. In this regard, IFN-γ and, to a lesser extent, IL-17 are crucial cytokines that activate macrophage fungicidal activity by promoting nitric oxide synthesis and reducing arginase activity [29]. In order to investigate the impact of LangDC1 depletion on the capacity of lung cells to inhibit in vitro fungal growth, pulmonary adherent cells were obtained 7 days after infection and cultured for 48 h with a *C. neoformans* (1 × 10^3^) yeast suspension. Figure 3B shows that adherent cells from LangDC1-depleted mice significantly inhibited yeast replication in vitro, evidenced by a reduction in the fold increase in CFU after culture, in comparison to that observed for *C. neoformans* cultured in medium alone or with cells from LangDC1-competent mice. It is noteworthy that pulmonary cells from LangDC1-competent mice were unable to control the in vitro growth of *C. neoformans* (Figure 3B), showing similar levels of CFU to yeast in medium alone. Moreover, a lower arginase-1 activity, measured by urea production, was detected in adherent cells from LangDC1-depleted mice compared with those from LangDC1-competent mice (Figure 3C). These in vitro results are supported by the arginase-1 and iNOS expression detected in situ by immunofluorescence of lung sections from infected untreated or DT-treated mice. Figure 3D shows that LangDC1-competent mice (*LangDTREGFP*, left panel) have an abundant infiltrate of arginase-1-positive cells with few detectable iNOS-reactive cells. In contrast, LangDC1-depleted mice (*LangDTREGFP* + DT, right panel) harbor scarce arginase-positive cells along with abundant iNOS-reactive cells in the lung parenchyma after 7 days of infection.

These data show that lung cells from LangDC1-depleted mice are better at controlling *C. neoformans* yeast replication and suggest a role for LangDC1 in inhibiting the antifungal activity of lung leucocytes in the early stage of cryptococcal infection.

## 4. Discussion

The primary target organ of *Cryptococcus neoformans*/*C*. *gattii* infection is the lung, and the early events of pulmonary immune response determine the outcome of fungal disease. To the best of our knowledge, this study provides the first evidence for the in vivo role of langerin-expressing DC1 pulmonary dendritic cells (LangDC1) in cryptococcosis. Our data show that LangDC1 modulate an early cytokine immune response and arginase-1/iNOS expression in the lung, promoting an environment susceptible to cryptococcal infection.

Intratracheal infection of C57BL/6 mice with *C. neoformans* strain 52D has been widely demonstrated to induce an allergic bronchopulmonary mycosis predisposing to chronic fungal infection [32,33]. The predominance of type 2 cytokines (IL-4, IL-13 and IL-5) in this model induces alternatively activated macrophages that are permissive to fungal replication due to an imbalance towards arginase-1 instead of iNOS activity [34]. In agreement with this, we observed that *LangDTREGFP* C57BL/6 transgenic mice were susceptible to lung infection, with abundant yeasts, inflammatory foci with neutrophil and eosinophil infiltration, type 2 cytokine production and predominance of arginase-1 activity and expression by lung cells at 7 days post infection.

Strikingly, our results reveal that in the absence of LangDC1, mice have better control of fungal infection, with lung cells producing IFN-γ and IL-17 even in the context of an exacerbated type 2 cytokine response. Therefore, type 1 and type 17 cell-mediated responses are probably effective at inducing an inflammatory environment able to reduce fungal proliferation, as demonstrated by a diminished fungal burden in tissue sections from LangDC1-depleted mice. In a previous study, pulmonary IFN-γ production in *C. neoformans*-infected C57/BL6 mice was shown to inhibit the growth of yeast in the lungs, even in the type 2 environment of allergic pulmonary mycosis [34]. Accordingly, in this study the lack of early IFN-γ production by pulmonary leukocytes from *C. neoformans*-infected LangDC1-competent (*LangDTREGFP*) mice might have impaired the fungicidal/fungistatic activity of lung cells.

Our data show that in the absence of LangDC1 cells, adherent lung cells from infected mice were more effective at inhibiting the growth of yeast in vitro and, at the same time, produced lower levels of arginase-1 than those from LangDC1-sufficient mice, suggesting a change in the activation profile of these cells. In this sense, IFN-γ and IL-17 cytokines are well-known for activating the anti-cryptococcal activity of macrophages by promoting nitric oxide synthesis and reducing arginase activity [4,29]. Accordingly, LangDC1-competent mice have a high fungal burden along with abundant arginase-1-expressing cells infiltrating the lung parenchyma and scarce iNOS-positive cells. Nevertheless, this scenario dramatically changes in the absence of LangDC1 cells since LangDC1-depleted mice are more resistant to fungal replication, which correlates with a decrease in arginase-1 and an increase in iNOS expression in lung tissue.

In fungal infections, lung DC1 are involved in a wide range of functions, from promoting innate antifungal immunity to reducing inflammation. For instance, DC1 have been reported to promote IFN-I-mediated protective immunity in *Histoplasma capsulatum*-infected mice [35]. Nevertheless, this DC subset has been shown to induce T regulatory differentiation in a pulmonary *Paracoccidioides brasiliensis* infection model [36] and to inhibit Th17 cell polarization in experimental invasive aspergillosis [37]. It is noteworthy that in the acute phases of pulmonary aspergillosis, IL-2 released by DC1 mitigated and controlled the pathogenicity of Th17 cells, representing a regulatory process specifically instructed by CD103+DC1 in response to the fungus [37].

In this regard, lung DC1 have been found to upregulate retinaldehyde dehydrogenase to induce Foxp3-expressing regulatory T (Treg) cells and promote airway tolerance to inhaled antigens [21]. Recent studies have demonstrated the central role of DC1 in suppressing pulmonary inflammation by inducing Treg, after phagocytosis of apoptotic cells [38].

In this study, in the absence of LangDC1, the anti-cryptococcal responses were modulated in the first week of infection, which suggests that these DC might be necessary in the regulation of innate immune cell functions. In this regard, it is known that early production of type 1 and type 17 cytokine pathways during cryptococcal infection may be mediated by γδT, NK, NKT or ILC innate cell populations, independently of adaptive Th1 or Th17 lymphocytes [2,39].

In summary, this study provides data showing that LangDC1 constitute a pulmonary subpopulation of DC that modulates the early anti-cryptococcal inflammatory response and the susceptibility to *C. neoformans* infection.

## Figures and Tables

**Figure 1 jof-08-00792-f001:**
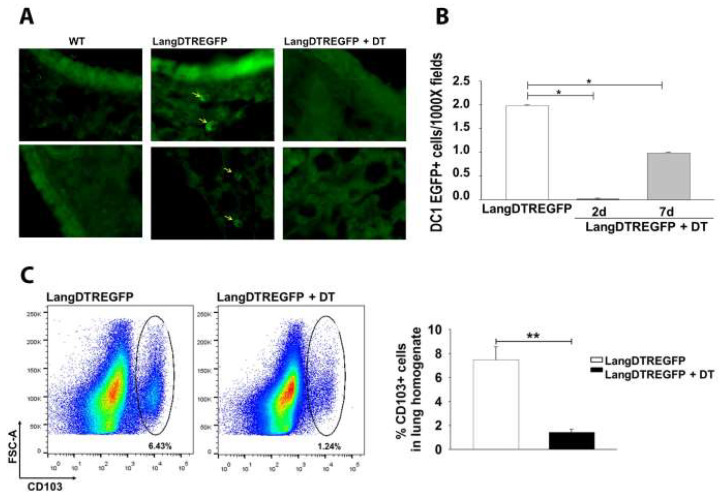
Conditional depletion of pulmonary langerin-expressing DC1 (LangDC1) after diphtheria toxin (DT) treatment of *LangDTREGFP* mice. (**A**) Representative microphotograph of immunofluorescence analysis of lung cryosections from untreated (saline) WT and *LangDTREGFP* mice or DT-treated (2 days post DT) *LangDTREGFP* animals. Arrows: langerin-expressing (EGFP+) cells. (**B**) Total number of EGFP+ cells counted per high-powered field within lung cryosections from untreated (white bars) or DT-treated LangDTREGFP mice, at 2 (black bars) or 7 (gray bars) days after DT administration. Data are expressed as mean ± SEM; bars represent data from two tissue samples from each animal analyzed at a magnification of 1000 X (three animals per group). * *p* ˂ 0.001. (**C**) Flow cytometry analysis showing representative dot plots of pulmonary DC1 (CD103+ gated on CD45+ CD11c+ live lung cells) after 4 days of saline or DT administration. The bar graph shows percentages of pulmonary DC1 (CD103+ CD45+ CD11C+ cells) from untreated (white bar) or DT-treated (black bar) *LangDTREGFP* mice. Data are expressed as mean ± SEM from triplicates of pooled lung cells (*n* = 3 animals per group) and representative of two independent experiments. ** *p* ˂ 0.026. All data were analyzed with Student’s *t*-test or ANOVA. DC1: conventional type 1 dendritic cells; DT: diphtheria toxin; WT: wild type.

**Figure 2 jof-08-00792-f002:**
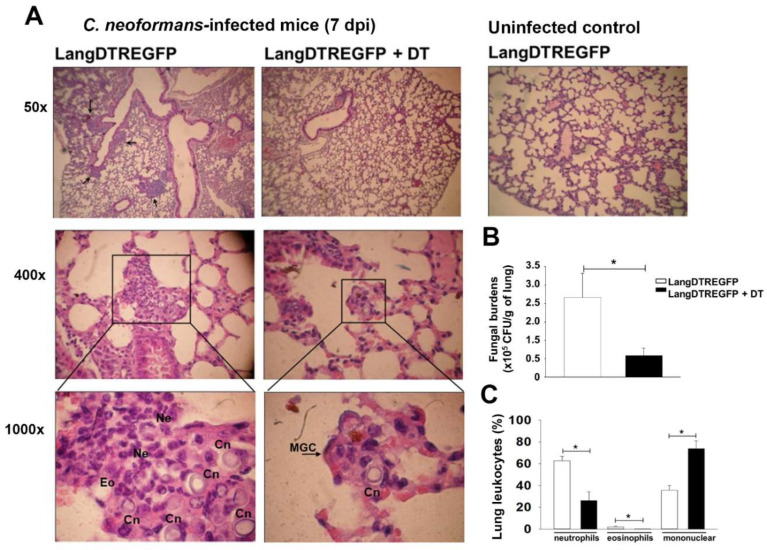
LangDC1-depleted mice better control pulmonary *C. neoformans* infection. (**A**) Histopathology (H&E) of lung sections from untreated or DT-treated *LangDTREGFP* mice, intratracheally infected with *C. neoformans* or uninfected (controls). Arrows (upper panels) and squares (middle and bottom panels) show inflammatory infiltrates and *C. neoformans* yeasts (Cn). Ne: neutrophils; Eo: eosinophils; MGC: multinucleated giant cells. (**B**) Fungal burden in lung homogenates (CFU/g) from DT-treated (black bars) or untreated (white bars) 7-day-infected *LangDTREGFP* mice. Data are mean ± SEM (*n* = 6 animals per group) of two independent experiments. * *p* < 0.037. (**C**) Percentages of neutrophils, eosinophils and mononuclear cells present in inflammatory infiltrates were counted per high-powered field within lung sections from DT-treated (black bars) or untreated (white bars) 7-day-infected *LangDTREGFP* mice. Data are mean ± SEM (*n* = 3 mice per group); two tissue samples from each animal were analyzed at a magnification of 1000X. * *p* ˂ 0.005. All data were analyzed with Student’s *t*-test or ANOVA. DT: diphtheria toxin; dpi: days post infection; CFU: colony-forming units.

**Figure 3 jof-08-00792-f003:**
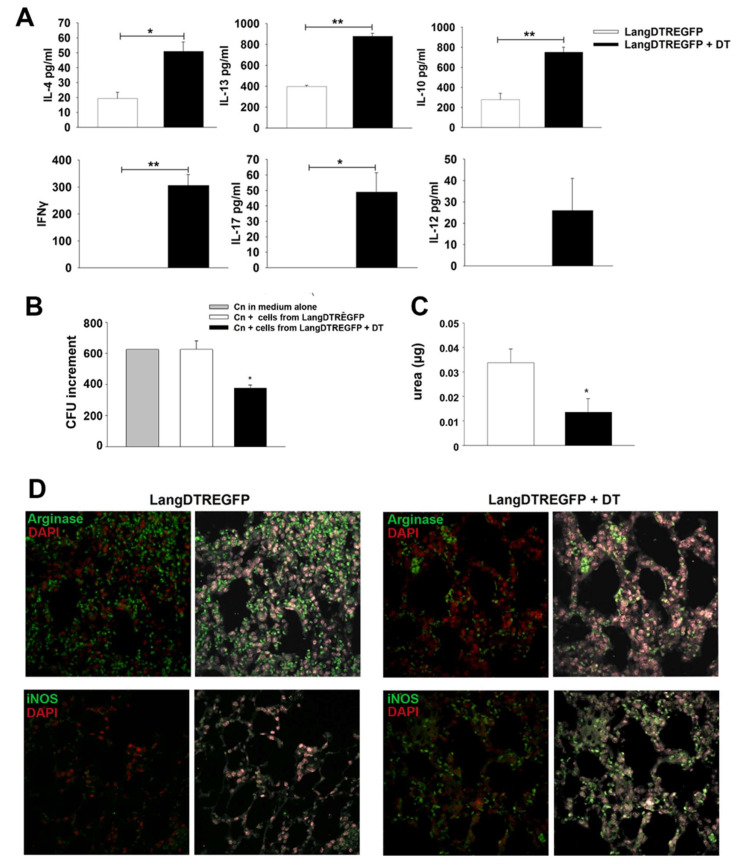
Ex vivo cytokine production, fungal growth inhibition and arginase-1 or iNOS expression by lung cells from *C. neoformans*-infected mice. (**A**) Cytokine production by lung cell suspensions obtained from DT-treated (black bars) or untreated (white bars) *LangDTREGFP* mice at 7 days post infection. Bars represent cytokine levels (ELISA) in 24 h culture supernatants. * *p* < 0.05; ** *p* < 0.005. (**B**) In vitro *C. neoformans* growth assay using a yeast suspension (1 × 10^3^) cultured in medium alone (gray bar) or in the presence of adherent lung cells from untreated (white bar) or DT-treated (black bar) 7-day-infected *LangDTREGFP* mice. Bars represent ratio of final vs initial CFU numbers in each culture condition. (**C**) Arginase activity from lysates of adherent cells cultured in (**B**), measured as urea production (µg of urea per µg of protein). * *p* < 0.05. Data are expressed as mean ± SEM. The data shown are pooled from two independent experiments (*n* = 6 animals per group; samples from each animal were analyzed in triplicate (**A**,**C**) or duplicate (**B**)). All data were analyzed with Student’s *t*-test or ANOVA. (**D**) Confocal microscopy of representative lung sections from 7-day-infected mice (*n* = 6) after immunostaining with fluorochrome-labeled specific antibodies to arginase-1 (upper panels) or iNOS (lower panels). 600× magnification. DT: diphtheria toxin; dpi: days post infection; CFU: colony-forming units.

## Data Availability

The data presented in this study are available in the article and Appendix A.

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
