# Peer review of "Pulmonary Conventional Type 1 Langerin-Expressing Dendritic Cells Play a Role in Impairing Early Protective Immune Response against Cryptococcus neoformans Infection in Mice"

_jof, 2022, doi:10.3390/jof8080792_

Round 1

Reviewer 1 Report

“Pulmonary conventional type 1 Langerin-expressing dendritic cells play a role in impairing early protective immune response against Cryptococcus neoformans infection in mice” is a resubmitted paper demonstrating that elimination of this DC population accelerates clearance of the fungus at least temporarily. The intriguing finding is that this specialized DC population appears to exaggerate the course of infection. Beyond that, the manuscript provides only correlative data. That is a significant issue for this manuscript.

1. The authors show that many cytokines both type 1 and type 2 are upregulated in the DC-depleted mice. I am not sure who is more important, IFN and IL-17 or IL-4, IL-13 and IL-10. That is not sorted out in this paper.

2. The analysis of the lung cell populations is only done microscopically which is less than an ideal way to look at populations. They should use flow cytometry which they are capable of doing.

3. They only look at 1 time point. Does this early time point reflective of later time points?

4. The use of pulmonary adherent cells is less than ideal. The authors need to find a better way to isolate macrophages. When adding pulmonary cells between groups who knows if the numbers of macrophages are similar. As evidenced in Figure 2 C there are more mononuclear cells in the lungs of deleted mice. What proportion of those mononuclear cells belongs to the macrophage lineage is not clear? Therefore, when adding an admixture of cells to a well and washing off the unbound, it becomes extremely dangerous to make conclusions unless one knows that the numbers are equal. The data in the supplement show differences in different populations. They can easily isolate F4/80+ cells and let them adhere and test their killing capacity. Moreover, frequency is insufficient. Need to know absolute cell numbers.

Author Response

Response to Reviewer 1

Pulmonary conventional type 1 Langerin-expressing dendritic cells play a role in impairing early protective immune response against Cryptococcus neoformans infection in mice” is a resubmitted paper demonstrating that elimination of this DC population accelerates clearance of the fungus at least temporarily. The intriguing finding is that this specialized DC population appears to exaggerate the course of infection. Beyond that, the manuscript provides only correlative data. That is a significant issue for this manuscript.

  1. The authors show that many cytokines both type 1 and type 2 are upregulated in the DC-depleted mice. I am not sure who is more important, IFN and IL-17 or IL-4, IL-13 and IL-10. That is not sorted out in this paper.

Thank you very much for the Reviewer’s suggestions.

In agreement with Reviewer’s comments regarding cytokine production in the lung, this work shows that DC1 would regulate type 1 and type 2 cytokine production. However, only in DC1-depleted mice do lung cells produce detectable levels of IFN-g, IL-12 e IL-17 , which are not measurable in DC1 competent mice. Regarding this, in the 3rd paragraph of the discussion we describe that pulmonary IFN-γ production in C. neoformans-infected C57/BL6 mice has been shown to inhibits the growth of yeast in lung even in the type-2 environment of the allergic pulmonary mycosis [33]. Interestingly, type 2 cytokines do not cross-regulate INF-g production.

The production of type 1 cytokines only occurs after DC1 depletion, so we think that the absence/presence of IFN-γ and IL-17 determines susceptibility or protection against infection, regardless of the type 2 environment in susceptible mice.

  1. The analysis of the lung cell populations is only done microscopically which is less than an ideal way to look at populations. They should use flow cytometry which they are capable of doing.

We include data from flow cytometry showing cell populations that correlates with finding of histology (supplementary figure 1)

  1. 3. They only look at 1 time point. Does this early time point reflective of later time points?

We are agreeing with the valuable data of a kinetic, but in this study we focused in early time of infection. In this same model of intratracheal infection with C. neoformas 52D strain in C57BL/6 mice  the type of immune response in the first week determines the susceptibility or resistance  against C. neoformans infection. Besides,  in this mouse strain,  the CFU peak in lungs occurs around day 7 pi and remains elevated for several weeks (Ref: Early Cytokine Production in Pulmonary Cryptococcus neoformans Infections Distinguishes Susceptible and Resistant MiceKathleen A. Hoag, Nancy E. Street, Gary B. Huffnagle, and Mary F. LipscombAm. J. Respir. Cell Mol. BioI. Vol. 13. pp, 481-495, 1995).

  1. The use of pulmonary adherent cells is less than ideal. The authors need to find a better way to isolate macrophages. When adding pulmonary cells between groups who knows if the numbers of macrophages are similar. As evidenced in Figure 2 C there are more mononuclear cells in the lungs of deleted mice. What proportion of those mononuclear cells belongs to the macrophage lineage is not clear? Therefore, when adding an admixture of cells to a well and washing off the unbound, it becomes extremely dangerous to make conclusions unless one knows that the numbers are equal. The data in the supplement show differences in different populations. They can easily isolate F4/80+ cells and let them adhere and test their killing capacity. Moreover, frequency is insufficient. Need to know absolute cell numbers.

Although in the experiments with adherent lung cells we did not determine the composition of these cells, it is well known that myeloid cells, mainly macrophages, express arginase-1 or iNOS after C. neoformans infection.  Accordingly, Hiten Madhani has recently been reported expression of arginase-1 in interstitial lung macrophages after infection with C. neoformans in C57BL/6 (Mycotalk 2022: https://www.youtube.com/watch?v=O7ZY7faLmaY&t=1656s). We were unable to obtain a good antibody and staining for double labelling of macrophages with arginase-1 or iNOS in immunofluorescence experiments.

In this revised version we include the absolute number of lung leukocytes analyzed by flow cytometry (Supplem Fig1 C).  Both the frequency and absolute number of F4/80 decrease in DC1-depleted animals, however it is clear that there are more iNOS+ cells in the lung parenchyma compared to the lungs of DC1-competent animals.

Unfortunately, we are currently not able to perform experiments to purify F4/80 cell due to LangEGFPDTR mice are not breeding any more in our Facility.  Performing new experiments would take a long time since we need to import this mouse strain again.

Reviewer 2 Report

The finding that langerin+DC1 impacted the fungal infection in lungs is very interesting. In figure S1, the & probably means p<0.001. The T cell infiltration is significantly higher in the langerin+ DC1 depleted mice. What is the frequency of the CD4/CD8+ T cell ratio in the infiltrated T cells in lung? Can authors provide intracellular FACS data of IFN and IL-17 producing T cells in the DT treated mice and controls, since these cytokines are important to activate macrophages. 

Author Response

Response to Reviewer 2

The finding that langerin+DC1 impacted the fungal infection in lungs is very interesting. In figure S1, the & probably means p<0.001.

Actually there was a mistake, the difference in the CD3 frequency correspond to p < 0.01 with ***. This was corrected in the revised Figure Suplem 1.

The T cell infiltration is significantly higher in the langerin+ DC1 depleted mice. What is the frequency of the CD4/CD8+ T cell ratio in the infiltrated T cells in lung? 

Can authors provide intracellular FACS data of IFN and IL-17 producing T cells in the DT treated mice and controls, since these cytokines are important to activate macrophages.

Unfortunately, we are currently not able to perform these Reviewer’s requests since LangEGFPDTR mice are not breeding any more in our Facility.  Performing new experiments would take a long time since we need to import this mouse strain again.

Reviewer 3 Report

This paper seeks to explore and understand more the role of DC1 in the pulmonary infection by Cryptococcus neoformans especially in the early stages. It shows the importance of LangDC1 in regulating the early response against C. neoformans, its function and role in and inhibiting mechanisms of fungal clearance.

In my opinion the paper is fit for the format of Short Communication, since it shows clear but few results. Nevertheless, the results are of importance since they show a new role for LangDC1 that could open the door for a broader and deeper understanding of the stages of infection.

The Introduction is well written and is mostly clear, rewriting the paragraph explaining the heterogeneity of DC subsets (lines 52-63) would be beneficial. Expanding a little more on the two subsets will highlight the importance of the paper more.

The Materials and Methods section is clear, but please specify which antibodies were used for the Flow Cytometry analysis.

The Results are well presented and explained, and show proof of concept for the conditional depletion of LangDC1 that is very essential for this study. Were the levels of EGFP for LangDTREGFP measured before and after DT treatment? Showing that is also important, not just CD103 levels.

Minor note for Figure 2, CGM is given for multinucleated giant cells instead of MGC both in the figure and legend.

For the discussion I would suggest that the authors try to address why they think in the presence LangDC1 the phenotype of allergic bronchopulmonary mycosis, is it the balance between Type 1 and Type 2 immune response? How do they plan on further studying that?

Overall, the paper is well written and clear and the conclusions are supported by the data.

Author Response

Response to Reviewer 3

Comments and Suggestions for Authors

This paper seeks to explore and understand more the role of DC1 in the pulmonary infection by Cryptococcus neoformans especially in the early stages. It shows the importance of LangDC1 in regulating the early response against C. neoformans, its function and role in and inhibiting mechanisms of fungal clearance.

In my opinion the paper is fit for the format of Short Communication, since it shows clear but few results. Nevertheless, the results are of importance since they show a new role for LangDC1 that could open the door for a broader and deeper understanding of the stages of infection.

The Introduction is well written and is mostly clear, rewriting the paragraph explaining the heterogeneity of DC subsets (lines 52-63) would be beneficial. Expanding a little more on the two subsets will highlight the importance of the paper more.

Regarding this suggestion, in paragraph 4 of Introduction we expanded description and function of DC1.

The Materials and Methods section is clear, but please specify which antibodies were used for the Flow Cytometry analysis.

According to Reviewer’s recommendation, we included description of antibodies in material and methods section

The Results are well presented and explained, and show proof of concept for the conditional depletion of LangDC1 that is very essential for this study. Were the levels of EGFP for LangDTREGFP measured before and after DT treatment? Showing that is also important, not just CD103 levels.

 The expression levels of GFP in lung cells using FACS was too low, so that’s why we had to measure CD103 expression. However, we were able to obtain a good image of GFP+ cells by immunofluorescence

Minor note for Figure 2, CGM is given for multinucleated giant cells instead of MGC both in the figure and legend.

This change was done in Fig 2

For the discussion I would suggest that the authors try to address why they think in the presence LangDC1 the phenotype of allergic bronchopulmonary mycosis, is it the balance between Type 1 and Type 2 immune response? How do they plan on further studying that?

The discussion was modified and we suggest that DC1 suppress early pulmonary response by inducing regulatory T cells that inhibit both type 1 and type 2 responses. However, in DC1-deficient mice the presence of type 1 cytokines with iNOS induction would be sufficient to inhibit fungal growth, even in an exacerbated type 2 environment. Studying the Treg response would be a good possibility in competent and DC1-depleted mice.

Overall, the paper is well written and clear and the conclusions are supported by the data.

Round 2

Reviewer 1 Report

The authors only address the importance of cytokines in the discussion rather than actually proving relative importance in vivo. This is inadequate.

The authors do show flow cytometry but they do not provide sufficient data regarding the numbers of each population and they do not discriminate among the various populations of immune cells. They show CD11b, CD11c but there are many populations that express those markers in the lung. They need better resolution and they need to calculate cell numbers not just proportions.